# (-)-Epigallocatechin-3-gallate Directly Binds Cyclophilin D: A Potential Mechanism for Mitochondrial Protection

**DOI:** 10.3390/molecules27248661

**Published:** 2022-12-07

**Authors:** Annan Wu, Jie Zhang, Quanhong Li, Xiaojun Liao, Chunyu Wang, Jing Zhao

**Affiliations:** 1College of Food Science and Nutritional Engineering, China Agricultural University, Beijing 100083, China; 2National Engineering Research Center for Fruit & Vegetable Processing, Beijing 100083, China; 3Beijing Key Laboratory for Food Non-Thermal Processing, Beijing 100083, China; 4Center for Biotechnology and Interdisciplinary Studies, Department of Chemistry and Chemical Biology, Rensselaer Polytechnic Institute, Troy, NY 12180, USA

**Keywords:** EGCG, CypD, interaction, mPTP

## Abstract

(1) Background: (-)-Epigallocatechin-3-gallate (EGCG) has been reported to improve mitochondrial function in cell models, while the underlying mechanism is not clear. Cyclophilin D (CypD) is a key protein that regulates mitochondrial permeability transition pore (mPTP) opening. (2) Methods: In this study, we found that EGCG directly binds to CypD and this interaction was investigated by surface plasmon resonance (SPR), nuclear magnetic resonance (NMR) and molecular dynamic (MD) simulation. (3) Results: SPR showed an affinity of 2.7 × 10^−5^ M. The binding sites of EGCG on CypD were mapped to three regions by 2D NMR titration, which are Region 1 (E23-V29), Region 2 (T89-G104) and Region 3 (G124-I133). Molecular docking showed binding interface consistent with 2D NMR titration. MD simulations revealed that at least two conformations of EGCG-CypD complex exist, one with E23, D27, L90 and V93 as the most contributed residues and E23, L5 and I133 for the other. The major driven force for EGCG-CypD binding are Van der Waals and electrostatic interactions. (4) Conclusions: These results provide the structural basis for EGCG-CypD interaction, which might be a potential mechanism of how EGCG protects mitochondrial functions.

## 1. Introduction

EGCG from green tea is a polyphenol compound famous for its antioxidant, anti-inflammatory, and anti-cancer activities [1], with benefits shown in many clinical, animal, and cellular experiments [2]. EGCG modulates cellular status through various molecular pathways [3]. EGCG was reported to improve mitochondrial function by inhibiting the opening of mPTP in cellular and animal models [4,5,6]. However, the molecular mechanism of EGCG improving mitochondrial function is not clear and remains to be investigated. Mitochondrion is the energy factory of the cell, providing energy for cellular activities through electron transport chains and oxidative phosphorylation coupling. The potential and ion concentration differences between the inner and outer membrane of the mitochondria need to be maintained to for proper mitochondrial function. Under oxidative stress or damage, the mitochondrial Ca^2+^ concentration increases dramatically, which in turn opens the mPTP, leading to changes in mitochondrial membrane permeability, causing mitochondrial swelling, impaired function, and necrosis [7].

To date, cyclophilin D is the only confirmed component of mPTP [8], which has been reported to be an important mPTP regulator. Pharmacological inhibition and gene knockout of CypD improved the symptoms of acute pancreatitis and reperfusion ischemia injury caused by mPTP opening [8,9]. Thus, CypD has been a hot target for the alleviation of mitochondrial dysfunction. Physical interactions between CypD and other proteins (p53, ATP synthase, Aβ…) are involved in the abnormal opening of mPTP [10]. Efficient blocking these interactions has been proved to prevent mitochondrial damage and cell necrosis[11,12]. Cyclosporin A (CsA), which is a well-established CypD inhibitor, can effectively inhibit mPTP opening by directly binding to CypD [13]. Considering the evidence that EGCG can inhibit mPTP opening, we speculate that EGCG may bind to CypD as well. Here, we demonstrated a direct interaction between EGCG and CypD by surface plasmon resonance (SPR) and identified the binding sites on CypD using 2D NMR titration. Molecular docking and dynamics simulations were carried out to uncover the details of EGCG and CypD interactions. Our work presents detailed information on the EGCG and CypD interaction, providing a fundamental structural basis for the use of EGCG as a potential CypD inhibitor regulating mPTP opening.

## 2. Materials and Methods

### 2.1. Materials

EGCG from green tea (458.4 Da, ≥95% purity) was ordered from Sigma-Aldrich (St. Louis, MO, USA). CypD was overexpressed and purified. Briefly, the CypD coding sequence was inserted downstream of the *Thermoanaerobactor tencongenesis* ribose-binding protein (RBP) gene of the pET41 plasmid, fused to a nucleotide sequence containing the HRV3C protease cleavage site, with a His tag at the 3′ end of the CypD gene. The plasmid was transformed into *E. coli* BL21 (DE3). The strain was induced to express the protein using IPTG. Then, protein was purified using a Ni^2+^-NTA column and lysed overnight using HRV3C protease (4 °C) and recovered using Ni^2+^-NTA and S75 size exclusion chromatography with a final purity of >95% as judged by SDS-PAGE. SPR CM5 sensor chip and amine coupling kit were obtained from GE Healthcare Bio-Sciences AB (Uppsala, Sweden). Deuterium oxide (D, 99.9%) was purchased from Cambridge Isotope Laboratories Inc (Tewksbury, MA, USA).

### 2.2. SPR Experiment

The SPR program of the BIAcore 3000 system was used to detect the binding strength of EGCG to CypD. First, CypD was immobilized onto the CM5 chip using EDC/NHS according to a standard amine coupling protocol. A 10μL solution of 0.1 mg/mL CypD was injected over the flow cell at 5 μL/min. Successful immobilization of CypD was confirmed by an ∼5000 resonance unit (RU) increase in the sensor chip. The first flow cell (control) was prepared without injection of CypD. After immobilization, the EGCG sample was diluted in running buffer (20 mM Tris–HCl, 150 mM NaCl, 1 mM TCEP, pH 7.2). Different dilutions of EGCG were injected at a flow rate of 30 μL/min for 3 min. Following sample injection, running buffer was passed over the sensor surface for a 3 min period for dissociation. The sensor surface was regenerated by a 20 μL injection of 10 mM glycine–HCl solution (pH = 2.25). The response was determined as a function of time (sensorgram) at 25 °C. Data was analyzed using Bioevaluation software, version 4.1.1 (GE Healthcare).

### 2.3. D NMR Titration

Two-dimensional NMR spectra of CypD were obtained at 25 °C on a Bruker 800 MHz NMR spectrometer equipped with a low-temperature probe. NMR data were processed and analyzed using Topspin 3.5pl7 and Sparky 3.115. The ^15^N labeled CypD was dissolved in 90/10% H_2_O/D_2_O with 25 mM NaCl, 50 mM Na_2_HPO_4_, 1 mM EDTA and 2 mM DTT at pH 6.8. A series of two-dimensional ^15^N-^1^H HSQC spectra were performed on 0.3 mM CypD samples by slowly adding different concentrations of EGCG. The chemical shift perturbation (CSP) of CypD for amide ^1^H and ^15^N chemical shifts were calculated using the equation $\mathrm{CSP}=\sqrt{100\times ∆H^{2}+∆N^{2}}$, where ∆H and ∆N are the differences between the chemical shifts of the free and bound forms of CypD, respectively.

### 2.4. Molecular Docking

Small molecule initial conformation was obtained from PubChem and protein initial conformation (PDB ID is 3r4g) was obtained from the Protein Data Bank (PDB) and use Autodock [14] for semi-flexible docking [15]. Lamarckian genetic algorithm (LGA) was used for the global search for 50 low energy conformations [16]. The lowest binding energy binding pattern obtained by docking matches the binding sites determined by two-dimensional nuclear magnetic titration. Then, the interaction pattern of EGCG with CypD amino acid residues was determined using LigPlot^+^ v.2.2 [17]. Pymol 2.2.0 was used to visualize the 3D structure of the complex [18].

### 2.5. Molecular Dynamics Simulation

CypD and EGCG in the binding mode described above was selected for molecular dynamics simulation. Gromacs 2021 was chosen for molecular dynamics simulation [19]. GAFF force field was used for small molecules [20]. Amberff99SBildn force field was used for protein [21], small molecule charges were optimized using ORAC 5.0 and Multiwfn calculation [22,23], then constructed small molecule force field file by Sobtop [24].First, protein and small molecule was added to a cubic box with 6 nm side length, filled with TIP3P water molecules and 5 Cl^−^ were added to balance system’s charge. Next, the system was energy optimized to convergence. Then, 200 ps NVT equilibrium was performed using V-rescale coupling [25] to stabilize the system temperature around 300 K; 1 ns NPT equilibrium was performed using Berendsen coupling [26] to stabilize the system pressure around 1 bar, and finally 1000 ns MD simulation was performed with a time step of 2 fs and the traces were saved every 10 ps. The LINCS algorithm was used to constrain the covalent bonds associated with heavy hydrogen atoms [27]. Van der Waals interaction and short-range electrostatic interaction were truncated at a radius of 12 Å. Long-range electrostatic interaction was calculated using the particle mesh Ewald (PME) method [28].

The root means square deviation (RMSD) of protein, the average distance between CypD and EGCG and the number of contacts <0.6 nm between CypD and EGCG were analyzed using gmx rms, gmx distance and gmx mindist provided by Gromacs 2021. Visual Molecular Dynamics program (VMD) was used to visualize the trajectories [29]. Gmx_MMPBSA [30] is a program based on Amber’s MMPBSA.py [31]. We can use it to calculate complex’s binding free energy and the binding contribution of each amino acid residue using the molecular mechanics Poisson-Boltzmann surface area (MMPBSA) method [32].

### 2.6. Statistical Analysis

The statistical analysis software GraphPad (San Diego, CA, USA) was used to analyze the data. A comparison of the means was ascertained by Duncan’s test at 5% level of significance using one-way analysis of variance (ANOVA).

## 3. Results

### 3.1. EGCG Binds to CypD by SPR

EGCG improves mitochondrial function by inhibiting the opening of mPTP [4,5,6]. Cyclophilin D is the only confirmed component of mPTP [8]. To test whether EGCG directly binds to CypD, CypD was immobilized on a CM5 SPR chip following a standard amine coupling protocol. Different concentrations of EGCG from 0.25 μM to 10 μM were flowed over the chip surface and the binding curves were recorded (Figure 1A). As shown in Figure 1B, RU values increased with the increase in the EGCG concentration, representing binding to CypD immobilized surface. The sets of binding curves fit well with the 1:1 Langmuir binding model, consistent with a monophasic binding process. The apparent on-rate (*k_on_*) of EGCG-CypD binding is 1.17 × 10^4^ M^−1^s^−1^, and the apparent off-rate (*k_off_*) is 0.316 s^−1^, illustrating a fast associate/dissociate interaction. The dissociation constant (*K_D_*) was obtained as 2.7 × 10^−5^ M (27 μΜ), indicating an overall moderate binding affinity.

### 3.2. Binding Sites Mapping by 2D NMR Titration

To investigate the binding sites of EGCG on CypD, a 2D ^15^N-^1^H HSQC NMR titration was carried out by gradually adding different concentrations of EGCG into ^15^N labeled CypD solution (Figure 2A). Significant chemical shift perturbations (CSPs) were observed after the addition of EGCG. CSPs were plotted against amino acid residue number in Figure 2C. Amino acid residues K91, H92, V93 and L98 exhibited the biggest CSPs, followed by V128, L24, H131, and V132. It is worth noticing that positively charged residues like lysine and histidine are favorable binding site for EGCG, indicating the presence of electrostatic interaction. Residues with hydrophobic side chain including leucine and valine are also highly involved, probably attributed to the hydrophobic interaction with EGCG ring region. Overall, three regions including E23-V29, T89-G104 and G124-I133 dominate the CypD-EGCG interaction. This result indicates that EGCG binds CypD possibly in a dynamic and multi-site mode, which is similar to a recent finding on EGCG-p53 interaction [33].

### 3.3. Binding Interface of EGCG-CypD Located by Molecular Docking

To further characterize the binding of CypD and EGCG, semi-flexible molecular docking analysis was performed with Autodock and 50 binding conformations with low binding energy were obtained by Lamarckian genetic algorithm [16], among which the lowest energy binding site is shown in Figure 3A. By LigPlot^+^ v.2.2 analysis, EGCG was found to interact with L90, V93, V97, H131, I133 through hydrophobic interaction; and forms hydrogen bonds with E23, K91, V128, G130. These interactions were consistent with the significant CSPs (E23-V29, T89-G104 and G124-I133) observed in NMR titration, again confirming the binding sites. CypD and EGCG binding surface was shown in Figure 3B. However, the molecular docking obtained only shows one conformation of EGCG and CypD interaction in vacuum. In order to investigate the EGCG-CypD binding dynamics in aqueous solution, we further performed molecular dynamics simulation based on this binding state.

### 3.4. MD Simulation Shows Two Modes of CypD-EGCG Binding

A 1 μs molecular dynamics simulation was performed on CypD-EGCG complex determined above using GAFF force field for EGCG [20] and Amberff99SBildn force field for CypD and explicit solvent [21]. To find out the binding details of EGCG and CypD, we calculated the RMSD of CypD, the average distance between CypD and EGCG and the number of contacts (if the distance between the atom of CypD and the atom of EGCG is less than 0.6 nm, it is considered as one contact) during simulation (Figure 4A). RMSD corresponds to the conformational changes of CypD from the starting structure during the simulation. The RMSD of CypD increased rapidly in the first 100 ns, while the average distance and contact number of CypD-EGCG did not change significantly, indicating that some conformational changes occurred in CypD to adapt to the aqueous environment. At around 500 ns, the average distance between CypD and EGCG dramatically increased while the number of contacts decreased, implying that the binding mode of EGCG and CypD changed at 500 ns.

To show the two binding modes between of EGCG and CypD, complex structures at 490 ns (conformation 1) and 510 ns (conformation 2) were shown in Figure 4B,C. MMPBSA method was used to calculate the binding free energy of individual amino acid residues. The contribution of individual amino acid residues was shown in Figure 4B,C. For conformation 1, residues E23, D27, L90 and V93 of CypD dominate the binding and the ring B and ring D of EGCG are highly involved. Residues V28, K91, H92, V97, V128, F129, G130 and H131 were also involved. For conformation 2, the top three contributing residues are E23, L5 and I133. Combined with NMR titration (Figure 2C), it seems that EGCG moved from a pocket formed by Region 1 and Region 2, to another pocket formed by Region 1 and Region 3. The contributing residues in conformation 2 was less than conformation 1(Figure 4B,C), consistent with a smaller number of contacts (Figure 4A). Overall, this conformation transition captured by MD is consistent with NMR showing CSPs in three different regions. The conformation of CypD did not show significant changes upon EGCG binding, probably due to that EGCG mainly interacts with the loop regions of CypD.

Binding free energies for the complexes were also calculated and shown in Table 1. The total binding free energies averaged over 400–500 ns (conformation 1) and 500–600 ns (conformation 2) are −29.3 kcal/mol and −22.52 kcal/mol, respectively, indicating a favorable interaction of both conformations. Electrostatic interaction dominated the binding energy with values o`f −57.34 kcal/mol and −47.04 kcal/mol. ΔG_gas_ and ΔG_solv_ are the binding free energy in vacuum and solvation energy, respectively. At 400–500 ns, ΔG_gas_ is −80.41 kcal/mol, contributed by Van der Waals (−23.07 kcal/mol) and Electrostatic interaction (−57.34 kcal/mol). ΔG_solv_ is 51.11 kcal/mol, mostly contributed by the solvation polarization energy of 53.93 kcal/mol, indicating that the solvation polarization energy is unfavorable to CypD and EGCG binding. At 500–600 ns, ΔG_gas_ is −57.74 kcal/mol and ΔG_solv_ is 35.21 kcal/mol, adding up to a total binding free energy of −22.52 kcal/mol. Overall, Van der Waals and Electrostatic interaction are the major driven force for EGCG and CypD binding.

## 4. Discussion

Cyclophilin D (CypD) is a key regulator of mPTP opening. This pathophysiological phenomenon is associated with the development of many human diseases, including acute pancreatitis, ischemia reperfusion injury, Aβ-associated neurodegeneration and others [9,34]. CypD interacts with proteins such as p53, Aβ, F0F1-ATP synthase, ANT, and PiC, which are all involved in the regulation of mPTP opening [10]. Accumulation of CypD and its complex with protein including p53, Aβ and others has been found in disease condition. Efficient interruption of these interactions has been proved to prevent cell death. EGCG has been reported to regulate enzyme activity by binding to an allosteric site [35] and interrupt protein aggregation [36]. In this study, a direct binding was identified for CypD and EGCG, a natural polyphenol compound form green tea. CypD binds different partners in different regions, thus the identification of binding site of EGCG on CypD is crucial. Combining 2D NMR titration and MD simulation, two CypD-EGCG complex conformations covering three binding sites (E23-V29, T89-G104, G124-I133) was discovered. It turns out that EGCG molecule tend to bind the highly dynamic loop region of CypD, in agreement with the findings of Fusco et al. [37] showing that EGCG likes random coils region in folded protein. Blocking mPTP opening through CypD inhibition could be a promising approach for mitochondria dysfunction. While numerous CypD inhibitors have been discovered to date, none have entered clinical practice, mostly owing to their high toxicity and unfavorable pharmacokinetics [38]. Here, we demonstrate that a non-toxic natural compound EGCG could be a potential CypD inhibitor, which offers important implications for the discovery of structural diverse CypD targeted molecules.

## 5. Conclusions

Our work demonstrates that EGCG directly interacts with CypD with a binding affinity of 27 μΜ. Three main regions on CypD, Region 1 (E23-V29), Region 2 (T89-G104) and Region 3 (G124-I133) were mapped as the binding interface by 2D NMR titration. Molecular docking and dynamic simulations revealed that at least two conformations exist for EGCG-CypD complex, with key interfacial residues consistent with those mapped from NMR CSPs. Our study provides the structural basis for EGCG-CypD interaction, which offers fundamental information for understanding how EGCG could interrupt CypD-partner interactions in mitochondrial damage. Our research also has important implications for the discovery of CypD inhibitor from natural products.

## Figures and Tables

**Figure 1 molecules-27-08661-f001:**
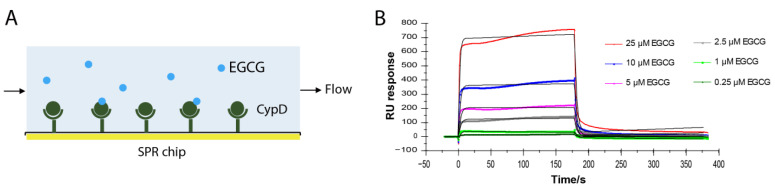
EGCG-CypD binding kinetics by SPR. (**A**) Schematic diagram for SPR experiment setup; (**B**) SPR sensorgrams of different concentrations of EGCG binding to CypD.

**Figure 2 molecules-27-08661-f002:**
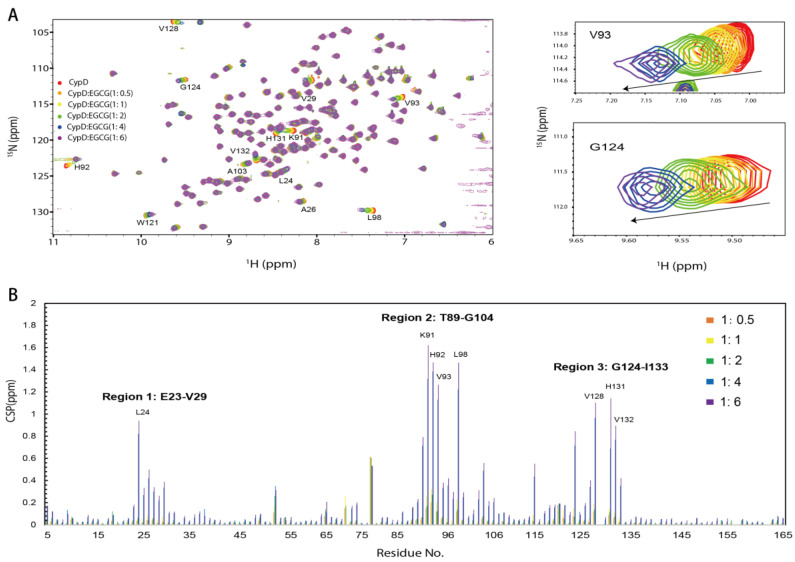
The binding sites of EGCG on CypD mapped by 2D NMR titration. (**A**) ^1^H-^15^N HSQC spectra of ^15^N CypD titrated by different ratios of EGCG (V93 and G124 were shown in magnified picture). (**B**) CSPs plotted against CypD residue numbers. (Amino acids with significant CSPs include Region 1: E23-V29, Region 2: T89-G104, Region 3: G124-I133).

**Figure 3 molecules-27-08661-f003:**
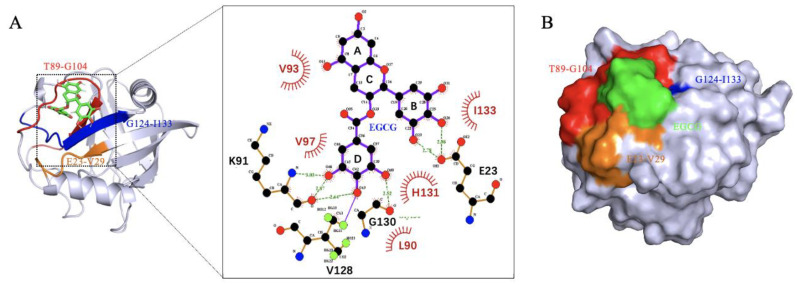
Molecular docking shows binding region consistent with 2D NMR. (**A**) EGCG binds CypD in the E23-V29, T89-G104, G124-I133 regions, forming hydrogen bonds with L90, V93, V97, H131, I133 and hydrophobic interactions with E23, K91, V128, G130; (**B**) Binding surface of EGCG and CypD.

**Figure 4 molecules-27-08661-f004:**
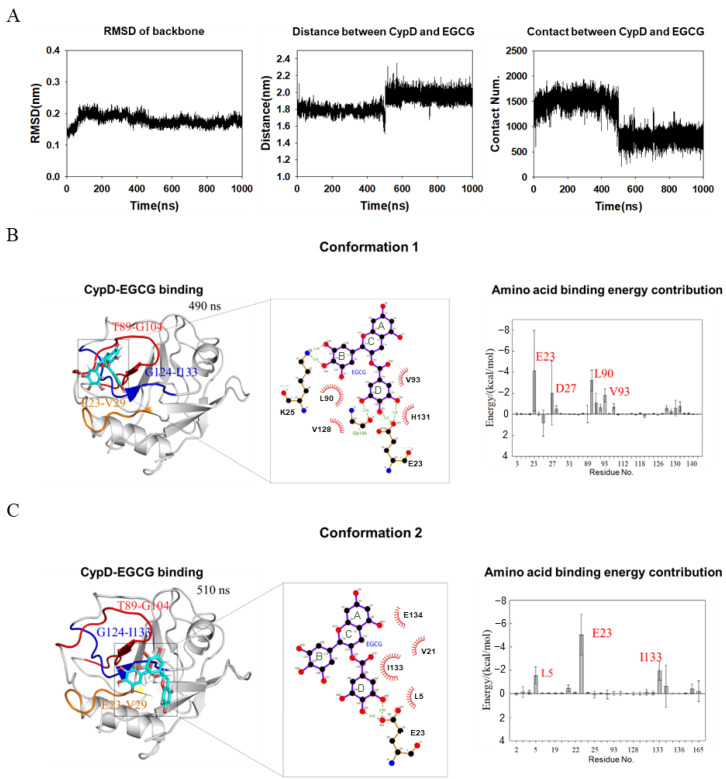
MD simulation shows two conformations of EGCG-CypD complex. (**A**) Stability of the EGCG-CypD system during molecular dynamics simulations; (**B**) EGCG-CypD complex conformation 1 and the energy contribution of interfacial residue; (**C**) EGCG-CypD complex conformation 2 and the energy contribution of interfacial residue.

**Table 1 molecules-27-08661-t001:** Binding energy of EGCG-CypD complex.

Energy Component	Average/(kcal/mol)	
	400–500 ns	500–600 ns
Van der Waals	−23.07	−10.70
Electrostatic	−57.34	−47.04
Polar solvation	53.93	37.30
Nonpolar solvation	−2.81	−2.08
ΔG_gas_	−80.41	−57.74
ΔG_solv_	51.11	35.21
Total	−29.30	−22.52

Notes: ΔG_gas_ = Van der Waals + Electrostatic, ΔG_solv_ = Polar solvation + Nonpolar solvation, Total = ΔG_gas_ + ΔG_solv_.

## Data Availability

Not applicable.

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
