# Peer review of "(-)-Epigallocatechin-3-gallate Directly Binds Cyclophilin D: A Potential Mechanism for Mitochondrial Protection"

_molecules, 2022, doi:10.3390/molecules27248661_

Round 1
Reviewer 1 Report
1. Although the topic is good but lacks appropriate discussion and analysis of results.
2. Cyclophilin D is important in regulating the mitochondrial permeability transition pore (PTP), interaction of (-)-Epigallocatechin-3-Gallate (EGCG) with this protein with high affinity can affect the function of the pore, what authors think of this. Any literature search.
3. Molecular docking or 2D NMR, which technique authors think is giving more detailed and reliable information regarding binding interaction. Why they must agree with each other.
4. What will be the mechanism of transport of (-)-Epigallocatechin-3-Gallate (EGCG) to mitochondria.
5. What about the conformational changes/ topology of Cyclophilin D upon the interaction with (-)-Epigallocatechin-3-Gallate (EGCG)
6. Line 40-“may contribute to mechanism of EGCG how improves mitochondrial functions” is very confusion authors need to clear their point of view
Author Response
Response to Reviewer’s comments
Dear Reviewer:
Thank you for your comments concerning our manuscript entitled “(-)-
Epigallocatechin-3-Gallate directly binds Cyclophilin D: a Potential
Mechanism for Mitochondrial Protection”. Those comments are all
valuable and very helpful for revising and improving our paper, as well as
the important guiding significance to our research. We have studied
comments carefully and have made correction which we hope to meet with
your approval. Revised portion are all highlighted in red in the paper. The
main corrections in the paper and the responds to the reviewer’s comments
are as following:
We’ve added more information in the introduction section. See lines 42-
48 in the revised MS.
1. Although the topic is good but lacks appropriate discussion and analysis
of results.
Response: Thanks for your constructive advice. We have added more
results analysis and discussion in the main text. Please see Lines 154-161,
Lines 208-213, Lines 236-256, Lines 263-266 in the modified main text.
We’ve also improved Figure 4 to show more details.
2. Cyclophilin D is important in regulating the mitochondrial permeability
transition pore (PTP), interaction of (-)-Epigallocatechin-3-Gallate (EGCG)
with this protein with high affinity can affect the function of the pore, what
authors think of this. Any literature search.
Response: Thanks for your valuable suggestion. EGCG could not readily
pass through inner membranes of mitochondria but may do so when inner
membranes integrity is compromised under oxidative stress or other
disease conditions (Weng Z. et al., 2014). Thus, EGCG will not affect
mPTP opening in normal cell. CypD is important in regulating mPTP
opening, which has been identified as a target for the treatment of acute
pancreatitis, Aβ-associated mitochondrial dysfunction, ischemiaassociated
necrosis, etc. Accumulation of CypD and its complex with
protein including p53, Aβ and others has been found in disease condition.
Efficient interruption of these interactions is proved to have positive effect
on mitochondrial protection. CypD binds different partners in different
regions, thus the identification of binding site of EGCG on CypD is crucial,
which we investigated in this paper.
References: Weng Z., et al., Green tea epigallocatechin gallate binds to
and inhibits respiratory complexes in swelling but not normal rat hepatic
mitochondria, Biochemical and Biophysical Research Communications,
2014, 443, 1097-1104
3. Molecular docking or 2D NMR, which technique authors think is giving
more detailed and reliable information regarding binding interaction. Why
they must agree with each other.
Response: Thanks for your question. 2D NMR is an experimental method
to map the binding site at atomic resolution according to chemical shift
perturbations, which is quite reliable. However, 2D NMR uses lots of
testing samples and is lack of visualization. Molecular docking and
simulation are good supplementary for NMR study, which not only
provides the binding site, but also visually shows the binding conformation
and simulates driven forces. Therefore, 2D NMR is more experimental and
molecular docking can provide more details. However, molecular docking
and simulation is highly relied on pocket selection, force fields selection
and software, which could be ambiguous without experimental data
support. Thus, a good consistency between experimental NMR data and
simulation data is an important indicator that our results obtained by
multiple methods are reliable.
4. What will be the mechanism of transport of (-)-Epigallocatechin-3-
Gallate (EGCG) to mitochondria.
Response: Thank you very much for your important comment. EGCG has
been reported to interacts with mitochondria in primary human umbilical
vein endothelial cells (HUVECs) (Piyaviriyakul S. et al., 2011), while
nothing has been reported on how EGCG is transported into mitochondria.
It is widely believed that the transport of EGCG across mitochondria
membrane is low in normal condition, while EGCG is more likely to be
transported when inner membranes integrity is compromised under
abnormal conditions such as oxidative stress (Weng Z., et al., 2014).
References: Piyaviriyakul S, Shimizu K, Asakawa T, et al. Antiangiogenic
activity and intracellular distribution of epigallocatechin-3-
gallate analogs. Biological & Pharmaceutical Bulletin, 2011, 34(3):396;
Weng Z., et al., Green tea epigallocatechin gallate binds to and inhibits
respiratory complexes in swelling but not normal rat hepatic mitochondria,
Biochemical and Biophysical Research Communications, 2014, 443,
1097-1104;
5. What about the conformational changes/ topology of Cyclophilin D upon
the interaction with (-)-Epigallocatechin-3-Gallate (EGCG)
Response: Thank you for your question. Our result didn’t show significant
conformation change of CypD upon EGCG binding, partially due to that
EGCG mainly binds to the unstructured loop region of CypD. We’ve added
this discussion in Lines 211-213 in the modified main text.
6. Line 40-“may contribute to mechanism of EGCG how improves
mitochondrial functions” is very confusion authors need to clear their point
of view
Response: Thank you very much for your proposal. We have corrected it
to the sentence below. See lines 263-266 in the modified main text.
Our study provides the structural basis for EGCG-CypD interaction, which
offers fundamental information for understanding how EGCG could
interrupt CypD-partner interactions in mitochondrial damage. Our research
also has important implications for the discovery of CypD inhibitor from
natural products.
Reviewer 2 Report
Manuscript entitled “(-)-Epigallocatechin-3-Gallate directly binds Cyclophilin D: a 2 Potential Mechanism for Mitochondrial Protection” by Annan Wu et al. was an enjoyable read with well-considered text and clearly presented high quality data. Provide novel insight about polyphenol compound of green tea (-)-Epigallocatechin-3-Gallate EGCG on physical interaction with cyclophilin D. Surface plasmon resonance (SPR), nuclear magnetic resonance (NMR) and molecular dynamic (MD) simulation data clearly show that EGCG directly binds to CypD. This potential interaction provides the molecular mechanism of EGCG to protect mitochondrial functions. Overall study is well planned and experimentally executed. Manuscript is well written and cited the supporting previous studies.
I have minor suggestions regarding figures font size, some of the fonts are not readable.
Author Response
Dear Reviewer,
Thank you for your comments. They are very helpful for improving our paper. We have increased the font size in the figures.
Round 2
Reviewer 1 Report
Authors almost answered all comments raised. But, I am still speculating that binding of EGCG can affect other functions in mitochondria and further investigation is needed in this direction, may be we will be able to have more information in future about transport and binding to other mitochondrial components.
still there are some typos and intext corrections needed to be revised.
Author Response
Thank you for your important comments. We’ll look into the effect of EGCG on other functions of mitochondria in the following studies. We went through the manuscript carefully and corrected all the typos.